# Recycling of Residual Polymers Reinforced with Natural Fibers as a Sustainable Alternative: A Review

**DOI:** 10.3390/polym13213612

**Published:** 2021-10-20

**Authors:** Natalia Fuentes Molina, Yoleimis Fragozo Brito, Jesús Manuel Polo Benavides

**Affiliations:** Environmental Engineering, Faculty of Engineering, Universidad de La Guajira, Riohacha 440003, Colombia; yoleimis@uniguajira.edu.co (Y.F.B.); jesusmanuel@uniguajira.edu.co (J.M.P.B.)

**Keywords:** residual polymers, natural fibers, biocomposites, environmental sustainability

## Abstract

The latest advances in green alternatives are being addressed with bio-based solutions, with uses and applications in new areas due to their wide potential, low cost, lightness, renewability, biodegradability, impact toughness, fatigue resistance, and other specific properties. Natural fibers are sustainable materials that have led researchers to test their viability as alternative reinforcements in residual polymers to meet required engineering specifications; therefore, it is essential to continue making progress in replacing conventional materials. This review is expected to provide an overview of the current scopes and future prospects of biocomposites from polymers reinforced with natural fibers with a focus on the following: i. recycling of residual polymers; ii. available natural fibers and their components in the context of engineering applications; iii. the behavior of the structural modifications of the natural fibers with the physical and chemical treatments in the matrix interaction as reinforcements of the residual polymers; and iv. applications for the development of innovative, efficient, and sustainable solutions for successful, environmentally responsible products.

## 1. Introduction

The use of more efficient biocomposites made from natural fibers and residual polymers in order to replace synthetic reinforcements has become a globally accepted alternative in various applications [1,2,3], due to their multiple advantages in some physical properties—mechanical and thermal [4,5,6,7]—added to the ecological contribution conferred by the use of residual materials with good yields.

In most cases, conventional materials reinforced with synthetic fibers such as glass, carbon, asbestos, beryllium, molybdenum, and aramid have unique advantages such as excellent stiffness, strength, and weight compared with bio-based materials [8]; however, they are used less and less due to high costs, inefficiency, high density, and unsustainable environments.

The need to innovate in sustainable biocomposites has grown steadily, conquering new engineering markets with simple and complex applications that are economically attractive and environmentally safe. Thus, the incursion of these materials has witnessed a stable expansion in use and volume with high performances [5,6,9], where careful selection of reinforcements allows any specific engineering requirement to be met.

In accordance with the above, natural fibers are widely used for their easy accessibility, renewability, nontoxicity, low-density (1.25–1.5 g/cm^3^), cost reduction, biodegradability, and satisfactory mechanical properties, making them an ecological alternative to replace glass and carbon fibers with densities of 2.54 y 2.1 g/cm^3^ [6,8,10,11], respectively, which means that they can be highly competitive, because they allow for the design of lightweight materials.

The use of residual polymers reinforced with natural fibers has been evaluated in different investigations with high yields and applications [12], indicating that the biocomposites are a function of the structure and nature of the reinforcing fibers; this is due to the high content of cellulose, which is hydrophilic in nature, affecting the interfacial bond with the polymeric matrix, which is hydrophobic [13,14].

It is important to mention that there are various investigations related to the use of reinforcements with natural fibers, where the effect of residual polymers on biocomposites is analyzed. Most of the authors consider that the drawbacks in the behavior of the union of the matrix and the reinforcements are caused by the hydrophilic nature of the fibers and the hydrophobic characteristics of the matrix [15,16], as mentioned earlier.

The physical, chemical, and biological treatment of natural fibers is one of the best ways to optimize the interaction between the fibers and the residual polymers within the biocomposite, reducing the hydroxyl functional groups OH^−^ present on the fiber surface and also increasing the surface roughness, therefore improving the interfacial interaction between the two and [12,17,18] determining the durability, rigidity, and resistance.

Faced with these expectations, this is a systematic review that aims to analyze the emerging area of residual polymeric materials with viable natural fibers; we determine the chemical composition of natural fibers, as well as the physical, mechanical, and thermal properties of the polymeric compounds formed with the fibers, to describe recycling in such a way that it contributes to the analysis of recovery and innovation strategies for the reuse of this material in the industrial sector.

## 2. Recycling of Residual Polymers

High production and increasing demand for polymeric materials causes considerable amounts of nonbiodegradable solid waste to accumulate in landfills, affecting the natural dynamics of ecosystems; as recycling is one of the most successful environmental management strategies with positive results that offers several application options, this section describes the technologies, the principles involved, and the challenges they imply in terms of efficiency, economic viability, and environmental solutions.

The recycling of residual polymers is carried out by different methods depending on the nature of the components and the application of the products obtained [19,20,21], the most common, as evidenced in Table 1, are as follows: i. mechanical recycling through crushing and/or grinding of polymers to reduce size—this is the most appropriate, with the least limitations and effect on the environment; ii. chemical recycling converts polymers into useful monomers in the petrochemical industry through solvolysis and hydrolysis; iii. thermal recycling of polymers by pyrolysis into new materials with energy use.

The environmental challenges of mechanical recycling of polymeric waste is an emerging field—that is, composites reinforced with natural fibers, described extensively in the present review, which lies in extracting natural fibers of low commercial value that are easy to obtain, to reinforce the matrices of these biodegradable products with various engineering applications [22,23].

### 2.1. Mechanical Recycling

Mechanical recycling is a method to reduce the size of polymeric materials into small fragments with sizes ranging between 20 and 200 mm [24]; polymeric compounds are crushed, ground, reprocessed, and combined with particles depending on the interest, and the resulting fractions of the process can be classified into different sizes using sieves in powders, fibers, or flakes [23,25]. They are used in the production of new products, with limitations due to their low density, mechanical strength, and interfacial adhesion; thus, it is necessary to apply treatments and use reinforcing fibers as mentioned below.

### 2.2. Chemical Recycling

The compounds present in the residual polymer disintegrate, recovering the monomers through the chemical depolymerization of the polymer chains [23] using different chemicals such as acids, bases, and solvents depending on the nature of the polymeric substrate. When organic and inorganic solvents are used, it is known as solvolysis; if water is used, it is known as hydrolysis [26,27], widely used in the petrochemical industry in the development of new alternative products and fuel, which is both economical and high-performance, with limitations due to the environmental impacts generated by the emissions and discharges of the process residues in the receiving environment.

### 2.3. Thermal Recycling

For energy recovery, thermal recycling is a process where heat is used to decompose the polymer into its different compounds, with variable temperatures ranging between 500 and 700 °C depending on the type of residual polymer [29,30]. The energy generated in the process is used most of the time, despite the drawbacks due to the by-products of ash and low-molecular-weight gases that are produced with the volatilization of some polymer compounds [28].

Thermal recycling can be performed by fluidized bed and pyrolysis processes, which are complex but their products are potentially useful and clean, similar to virgin products [24,31]. The fluidized bed process passes a rapid stream of hot air through silica sand to break down the composites present in polymers at low temperatures [31] and the pyrolysis process heats the composite materials present in the polymer in the absence of oxygen, which decompose, producing oil, gas, fibers, and fillers [20,24].

Advances in research on the recycling of polymers reinforced with natural fibers have been responding to the growing concerns of environmental management, making it an area of interest for the transformation of polymeric waste into commercially viable products; however, it is necessary to improve knowledge of recovery rates, the useful life of products, large-scale recycling, increased compatibility with the environment, and ensure that they are socially safe.

## 3. Natural Fibers as a Sustainable Alternative to Reinforce Residual Polymers

Natural fibers are flexible filaments extracted from renewable sources, with complex properties due to the wide variations in the chemical and structural composition of cellulose, lignin, hemicellulose, fatty acids, pectins, and other structures linked by intermolecular hydrogen bonds and forces of Van der Waals forming microfibrils with parallel arrangement. They are used in different fields of engineering as reinforcing elements that provides resistance to tension and bending, rigidity, and modulus of elasticity [13,32,33] forming a remarkable bioengineering material that is of interest for their multiple uses and applications across a wide range of products with pleasing value; they differ from the initial materials, which is related to the nature of the fibers, crystallinity, and insolubility; however, the elaboration of these composite materials in the melt process can present drawbacks such as the viscosity, which can become very high, especially when the fiber content is greater than 50% by weight; Mazzanti et al. [34] stated in their research that if the capillary is too narrow, the pressure can become too high and the use of lubricants promotes the sliding of the wall, which can alter the viscosity measurements if researchers do not account for wall slip appropriately [35,36].

In natural fibers, the compounds as a whole are held together by the collective function of cellulose, hemicelluloses, lignin, and pectin as matrix; on average, fibers are mainly composed of cellulose (60 to 80%), lignin (5 to 20%), and around 20% moisture [9]; these percentages can vary according to the type and origin of the fibers [32]. Next, the chemical composition of natural fibers are analyzed in Table 2 using data from different investigations since their understanding allows us to predict and infer their thermal and mechanical performance [33], coupled with the economic and environmental problems that can be solved with this alternative when they are linked to new bio-based compounds.

### 3.1. Cellulose

Cellulose is the major structural component of natural fibers. It is a linear biopolymer composed exclusively of β-glucose molecules linked together by 1,4 bonds that impart good resistance, rigidity, structural stability, porosity, and elasticity to the fiber [9]. This polysaccharide is composed of crystalline and amorphous microfibrils helically aligned along the fiber [11] and is resistant to hydrolysis and oxidizing agents, which can partially degrade in strong acid catalyzed media [37].

### 3.2. Hemicellulose 

Hemicellulose is a structure of straight branched chains composed of the various, most-abundant polysaccharides of lower molecular weight that form a branched chain, including β-glucose, mannose, galactose, or xylose, and the acetyl groups that contain side groups, giving rise to their noncrystalline nature, with a medium degree of polymerization through the covalent bonding of these compounds and by ionic and hydrophilic interactions; it is responsible for the thermal and biological degradation of the fiber through moisture absorption [38], which can be easily hydrolyzed by dilute acids and bases [11].

### 3.3. Lignin

Lignin is a key component in the fiber structure due to the complexity of its molecular structure. It is an amorphous and cross-linked three-dimensional polymer that acts as a natural binder of the individual fibers—filling the spaces between the pectin, hemicellulose, and cellulose [13,51]—composed of an irregular matrix of linked hydroxy and methoxy substituted phenylpropane units [9], which is responsible for stiffness, producing a structure resistant to impact and stress [42]. It is hydrophobic, resists acid hydrolysis, is soluble in hot alkali, easily oxidizes, and is responsible for radiation degradation despite being thermally stable.

### 3.4. Pectin

Pectin is an anionic polysaccharide complex that provides flexibility to the fibers since its structure is highly branched [37]; it is a structural acid heteropolysaccharide that is composed of modified glucuronic acid and rhamnose residues. The structural integrity of the plant is enhanced by the pectin chains that often cross-link with calcium ions [9].

Finally, there are extractives and ashes in a lower proportion, with great influence on the properties and processing of natural fibers, since they act as protectors; inhibit the attack of acids; and belong to different classes of organic and inorganic chemical compounds, respectively, which are extracted by washing the fibers with water or organic solvents prior to their binding in the polymeric biocomposite.

## 4. Properties of Natural Fibers and Their Wide Potential as Reinforcements in Residual Polymers

Natural fibers have a number of physical and chemical properties depending on the content of cellulose, lignin, hemicellulose, and pectin, which make then excellent materials along with their renewability and biodegradability, being a common practice to eliminate lignin and pectin to improve the reinforcing effect of natural fibers in biocomposites [52].

Natural fibers when used as reinforcements in biocomposites, in addition to representing environmental benefits, reduction in energy consumption, insulation properties and acoustic absorption [53,54], also have essential mechanical properties, as evidenced in Table 3, with average variations ranging from 1.25–1.5 g/cm^3^ for density, 320–520 Mpa for tensile strength, 22–48 Gpa for tensile modulus, and 7–25% for elongation before break; additionally, the investigations of Nurazzi et al. [10], Nagaraj et al. [55], and others related flexural strength, modulus of elasticity, thickness swelling, and water absorption as important properties when evaluating them as polymeric reinforcements.

The mechanical properties in natural fibers are lower than in synthetic fibers; they can be improved or equalized by surface modification techniques [1] (as presented below), in addition to the low density, which is one of the properties that makes them more attractive for different purposes and engineering applications, such as in construction, aeronautics, and automobiles [64,65]. On the other hand, it should be noted that the resistance of the fiber depends on the load imposed on the fiber, the weight ratio of the fiber, the cultivation process, the manufacturing or modification process, and the manufacturing methods of the reinforced polymeric matrices [66].

## 5. Treatments to Natural Fibers to Optimize the Interaction with Residual Polymers of Biocomposts

The use of natural fibers in residual polymers presents restrictions, due to limitations in thermal stability, interfacial adhesion, wetting, porosity, dispersion, and distribution in the matrix [18,67,68], making the physical and chemical modifications necessary to correctly transfer the load from the polymeric matrix to the natural fiber.

Given the incompatibility between the hydrophilic nature of the fibers and the hydrophobic character of the polymeric matrices [10,69], recent research has focused on the development of treatments to improve the dimensional stability of residual polymers in humid conditions (Table 4), which undergoes efforts by the expansion of the fiber, as there are no links between both structures, presenting shrinkage; propagation of microcracks; and deterioration of physical, chemical, and mechanical properties. 

The selection of physical and chemical treatments depends on the properties that can be generated in the residual polymeric matrices such as rigidity and resistance, degree of compatibility, functionality, acidity, and homogeneity [81,82,83,84] in addition to the toxicity, cost, handling, and availability; the best known are the alkaline, silian, acetylation, benzoylation, potassium permanganate, hydrogen peroxide, and stearic acid methods.

### 5.1. Physical Treatments

Physical modifications are made primarily to separate natural fibers into discrete filaments, altering the structural and surface properties of the fibers, to increase the mechanical bond to the residual polymers [9,85]; some physical methods, such as surface fibrillation, electric shock, and steam explosion, are used to change the properties by surface oxidation [42].

In plasma treatment, the fiber surface is exposed to high-voltage conditions to ionize the gas and use it as a plasma under vacuum conditions [79], thus improving the surface resistance of the fiber cross-linking the surface by introducing free radicals [9] managing to increase the surface tension of nonporous substrates. Corona treatment carried out in a discharge reactor modifies the oxidation of the fiber surface to improve the interface between the hydrophilic fiber and the hydrophobic matrix [80] and improves mechanical resistance by changing the surface energy and acidity of the fiber surface [86].

### 5.2. Chemical Treatments

The alkalization process known as mercerization causes fibrillation and increases the available area; it is effective for increasing the adhesion strength with the polymeric matrix, by modifying the chemical structure of the compounds present in natural fibers; it is useful for improving the roughness of the surface by altering the hydronium ions H^+^ and hydroxyl OH^−^ in the structure, where hydronium ions are broken and hydroxyl ions are more active, facilitating compatibility when dissolving amorphous components [18,50,68]. Modifications based on previous research by Senthilkumar et al. [69], Senthilkumar et al. [70], and Debeli et al. [68] are a function of the concentration of the solution, contact time of the fibers, temperature, and the type of alkaline compatibility by dissolving amorphous components (potassium hydroxide KOH, lithium hydroxide LiOH, and sodium hydroxide NaOH) during treatment, presenting a better degree of crystallinity and orientation of the fiber, which causes the thermal and mechanical properties of the reinforced polymeric matrices to change.

Silane is another coupling agent, responsible for improving mechanical properties to achieve thermal stability in the residual polymeric matrix. Silane molecules form covalent bonds through siloxane bridges, improving its interfacial adhesion; the effectiveness is due to the formation of Silanol (Si^−^OH) groups that form strong bonds with the hydroxyl groups ^−^OH of the fibers [4,10,74].

The remaining Silanol condenses with the adjacent OH ions, the hydrophobic polymerized silane thus formed can be attached to the polymer matrix by van der Waals forces, resulting—according to Thyavihalli et al. [6], Atiqah et al. [72], and Sepe et al. [73]—a better interface between natural fibers and residual polymers. 

Acetylation is another method of chemical transformation known to modify the surface of natural fibers and make them more hydrophobic, which consists of a substitution reaction of the hydroxyl ions ^−^OH of the polymer by acetyl (CH_3_OH), exhibiting a more adhesive behavior within the polymeric matrix. There are few investigations of this treatment; authors such as Aaliya et al. [1] and Gowthaman et al. [43] mentioned that it improves the dimensional stability of the compounds and imparts a rough surface with fewer voids, providing a strong mechanical enclave with the reinforced polymers.

Permanganate modifications also reduce the hydrophilic tendency of the fiber due to copolymerization by grafting of the cellulose matrix induced by the manganese ion Mn^2+^, which is highly reactive as an oxidizing agent that reacts with the hydroxyl ions ^−^OH of the cellulose and form manganese, and is used in Roy et al. [4], Moonart [77], Gowthaman et al. [43], and Bordoloi et al. [60] to modify the interfacial interaction and the performance of the molecular structure of fibrous materials in the polymeric matrix.

Peroxides break down to form free radicals that react with the hydrogen group of cellulose fibers and the polymeric matrix. The research of Thyavihalli et al. [6], Maslowski et al. [75], and Kumar et al. [63] perform peroxide modifications of natural fibers after alkalization and prior to benzoylation as a complementary treatment, since it reduces the tendency to absorb moisture by part of the fiber and, therefore, improves thermal stability.

Finally, benzoylation is a process where hydroxyl ions ^−^OH are replaced by benzoyl groups and bind to the cellulose structure; grafting on the polymeric matrix is induced by the peroxide that adheres to the surface of the fiber. Peroxide-initiated free radicals react with fiber and matrix hydroxyl. Therefore, peroxide treatment—based on the results of Thyavihalli et al. [6], Ali et al. [76], and Kumar et al. [63]—improves reinforcement–polymer interface properties, slows down the absorption of moisture, and improves thermal stability by reducing the hydrophilic nature of the fibers, increasing the resistance of the compounds, and providing greater thermal stability.

## 6. Properties of Polymers Reinforced with Natural Fibers

Residual polymers reinforced with natural fibers exhibit appreciable physical, chemical, and mechanical properties necessary for different purposes or applications [11,87,88]. The final properties of the materials are described below depending on the type of natural fiber, the percentages of addition, the surface treatment of the fiber, and the processing of the biocomposites [6,42].

Density in reinforced polymeric materials has a directly proportional relationship with mechanical properties—the denser the material, the stronger and more durable it becomes [88]. They also allow to improve the interfacial adhesion between the polymeric matrix and the natural fibrous reinforcement [1]; positive variations in density improve the modulus of elasticity, static bending strength, and modulus of rupture for compressive strength [89].

Hardness is a characteristic of reinforced polymeric materials that depends on the cohesion of the fibers and their structure, presenting greater difficulty to be penetrated by other bodies. The investigations of Roy et al. [4] and Wahab et al. [89] reported a directly proportional relationship with the density of the material. Significant increases in the hardness of reinforced polymers occur with high percentages of natural fibers present in the matrix [10].

Tensile strength, Young’s modulus, and elongation at break vary positively based on the percentages of reinforced natural fiber and largely determine the tensile properties of polymeric compounds according to Azad et al. [87] and Azammi et al. [32]; allowing us to determine, in addition to the resistance to breakage, the structural design of the surface of the matrix [88].

Residual polymers exhibit better flexural strength when bent or curved in their longitudinal direction under transverse loads and when they are reinforced with natural fibers; however, Aaliya et al. [1] and Ramamoorthy et al. [90] mentioned that Young’s modulus and the moment of inertia of the material show minimal changes along the transverse direction.

Impact resistance is another of the properties that has made it possible to characterize residual polymers due to the tenacity of its structure once it has been reinforced, allowing us to gain knowledge of the axial load that the materials can support in a small segment [88]. Natural fibers have a low impact resistance, where the factors that affect the impact resistance of structures are a function of the type of fiber, particle size, adhesion, interfacial test, and matrix condition [1,90].

To improve impact resistance, hardeners or impact modifiers are used in the production of biocomposites. The investigations of Khui et al. [91] and Meekum and Khongrit [92] used hardeners to improve the interfacial bond between the fiber and matrix, finding an increase in the values of impact resistance and elongation at break. These additions are not only used as a hardener, but also as compatibilizers that increase the tensile properties of materials [93].

Mazzanti et al. [94] used rubber as a hardener in the elaboration of polymeric matrices based on polypropylene reinforced with wood, with different percentages of addition and processing method, and found that to provide a significant improvement in the toughness and ductility of the compound, a minimum amount of 20% by weight of hardener is required.

In general terms, according to the research of [10,18,33,40,44,50], the most common fibers in the reinforcements of polymeric matrices are Kenaf, Coco, Cotton, Hemp, and Sisal, presented in Table 5: i. Coconut or coir fibers are attractive, for their durability compared to most natural fibers; ii. cotton fibers have excellent absorbency and account for 46% of world fiber production; iii. hemp fibers have excellent mechanical strength and Young’s modulus; iv. Kenaf fibers have low density and good specific mechanical properties; v. Sisal has high tensile strength and strength, resistance to abrasion, resistance to salt water, and resistance to attacks with acids and alkalis.

## 7. Feasible Applications for the Recycling of Reinforced Residual Polymers as Innovative and Sustainable Solutions

Various researches have attempted to explore the applications of residual polymers reinforced with natural fibers, with different approaches for the incorporation of materials in various sectors of industrial production; the present review proposes to broaden the understanding of the potential of polymeric residues reinforced with different fibrous materials that allows them to substitute conventional materials, as evidenced in Table 5.

In recent decades, residual polymers reinforced with natural fibers have presented a sustainable future in the areas of aeronautics, automotive, construction, biomedicine, and infrastructure, among others, with a wide variety of materials, which sometimes include improvements when compared with materials commonly used. Improvements in the performance and durability of materials have driven the growth of applications with polymeric composites reinforced with natural fibers in new potential areas, turning them into sustainable renewable resources to partially or totally replace synthetic materials.

Due to their excellent properties, natural fibers have quickly succeeded in replacing synthetic fibers, showing adequate tensile strength and flexural modulus compared with glass and aramid fibers [9] that makes them attractive for use in various fields of engineering, construction, aeronautics, medicine, sports, and footwear, among others. Composite materials have a wide field of application, where three main areas can be highlighted: automotive, aeronautics, and construction:

The automotive industry has significant potential for biopolymers, due to the demand for lightweight and environmentally friendly materials; studies indicate that natural fiber composites can contribute to a 20% cost reduction and a 30% weight reduction of an auto part [64]. German car companies (BMW, Mercedes, Volkswagen, Audi Group, Ford, Opel, and Daimler Chrysler) use natural fibers to make door panels, roof panels, trunk liners, seat backs, sound insulation panels, and linings, among others [66].

In aerospace applications, the main purpose is to reduce the weight of the structure [96]; approximately 50% of the components of the airspace are made of biopolymers reinforced with natural fibers [99].

Building materials are another important application of natural-fiber-reinforced composites; it is the sector with the highest consumption of natural resources and is expected to increase due to the housing deficit worldwide [66]. The application of biocomposites, from the recycling of polymeric waste, seems to be a solution to the problem of handling such waste [65].

## 8. Challenges of Polymeric Reinforcements in the Context of Engineering Applications

Polymeric reinforcements with natural fibers face certain challenges in terms of moisture absorption, adhesion, fire resistance, and weather dependence, which vary in consistency [100] in the context of engineering applications in the development of the efficient and sustainable solutions presented below:

High moisture absorption is the main drawback of polymeric reinforcements [33,101]. This phenomenon reduces the interfacial bond between polymers and natural fibers, causing detrimental effects on the final properties of the biocomposites, because as the composite material absorbs water, it becomes more susceptible to attack by microorganisms and the decomposition of the structure [8,9,56].

Biocomposites are less rigid, which is a critical aspect for the development of technological structures within different industries, especially the construction area. Although natural fiber composites have a high resistance, their stiffness is lower than that of any synthetic compound, which is necessary to design systems used to address the drawbacks in most cases [102,103].

Thermal stability is restricted in biocomposites, due to the use of natural fibers. Since they cannot withstand high temperatures, they begin to degrade and shrink when exposed to heating conditions; therefore, it is necessary to establish time intervals and temperature ranges to avoid facilitating its handling [13,87].

## 9. Final Considerations

In the attempt to promote the recycling of polymeric waste, strategies of valorization and innovation in different sectors were analyzed through the understanding of the physical, chemical, and mechanical behavior of the reinforcements with natural fibers, achieving the following conclusions:

The natural fibers that best respond to physical and mechanical properties as reinforcements in biocomposites with residual polymers are kenaf, cotton, coconut, hemp, and sisal; combined with its easy availability, low cost, durability, and absorbency, it allows us to find answers to specific requirements for performance in different areas of application.

The treatments most applied to natural fibers to improve the effectiveness of residual polymers are the alkaline and silian methods, which manage to modify the surface of the fiber, allowing us to improve the thermal stability, the interfacial bond, and the porosity, thus enhancing the mechanical properties since the charge transfer of the biocomposite is improved.

The best engineering applications in the design of materials reinforced with biopolymers are the automotive, construction, and aeronautical industries, with products of low density and sustainability, and a high demand—as the knowledge of composites and current market trends increases—to decrease the high demands on natural resources.

Unquestionably, the innovations of polymeric reinforcements with natural fibers leave an open path, due to the high level of acceptance of these materials, seeking to improve their performance and value to promote new areas of use and application.

## Figures and Tables

**Table 1 polymers-13-03612-t001:** Recycling methods, techniques, reuse, and drawbacks.

Recycling Method	Techniques and Processes	Reuse	Drawbacks	Authors
Mechanical	Crushing and/or grinding to reduce the size of polymer waste	Raw material to produce compounds for the same or other applications.	Limitations in its use due to the low density, interfacial adhesion, and mechanical resistance of the resulting fibers.	[22,24,25]
Chemical	Chemical degradation of polymeric residues into basic chemicals.	Raw materials for petrochemical industries. Alternative fuels.	Water and air pollution problems; dangerous for the health of recyclers.	[22,26,27]
Thermal	Recovery by pyrolysis and fluidized bed.	Energy, cement production.	Problems of atmospheric pollution due to the emission of low-molecular-weight gases.	[22,28,29]

**Table 2 polymers-13-03612-t002:** Chemical composition of some natural fibers.

Natural Fibers	Scientific Name	Cellulose (%)	Hemicellulose (%)	Lignin (%)	Extractive (%)	Ashes (%)	Author, Year
Sugar cane	*Saccharum officinarum*	46.6–45.1	25.5–25.0	20.7–14.1	29.4–2.7	8.0–2.6	[13,39]
Pineapple	*Ananas comosus*	81.2–45.0	50.0–12.3	30.0–3.4	_	_	[2,40]
Banana	*Musa paradisiaca*	60.0–10.0	19.0–16.0	19.0–5.0	9.6–2.0	11.0–1.2	[40,41]
Hemp	*Cannabis sativa*	72.0–68.0	15.0–10.0	10.0–3.0	_	5.8–2.3	[11,42]
Coconut	*Cocos nucifera*	53.0–43.0	14.7–1.0	45.0–38.4	_	_	[42,43,44]
Jute	*Corchorus capsularis*	72.0–60.0	22.1–13.0	15.9–13.0	_	3.0–2.5	[11,42]
Oil palm	*Elaeis guineensis*	45.0–28.2	18.8–12.7	49.5–9.4	7.13–2.0		[45,46]
Wheat	*Triticum*	43.2–60.5	34.1–20.8	22.0–9.0	_	5.7–5.6	[14,42]
Kenaf	*Hibiscus cannabinus*	65.7–63.5	17.6–15.3	21.6–12.7	4.0–2.0	2.2–1.0	[13,47]
Flax	*Linum usitatissimum*	81.0–70.0	20.6–16.7	10.0–3.0	_	_	[11,43]
Ramie	*Boehmeria nivea*	73.0–69.8	14.0–9.6	3.9–1.6	_	_	[44,47,48]
Sisal	*Agave sisalana*	75.0–65.0	13.9–10.0	10.0–7.6	_	1.0–0.4	[42,49,50]

**Table 3 polymers-13-03612-t003:** Physical and mechanical properties of some natural fibers.

Natural Fibers	Density (g/cm^3^)	Endurance Traction (Mpa)	Tension Module (Gpa)	Elongation at Break (%)	Author, Year
Wood	0.50–1.4	130–64	70–7	_	[56,57]
Flax	1.5–1.4	650–250	70–27	3.2–2.3	[6,8,10,56]
Hemp	1.6–1.4	690–630	70.0–45	3.0–1.6	[43,58,59]
Jute	1.5–1.3	773–325	55–26	2.5–1.5	[8,56,60]
Coconut	1.5–1.2	180–146	6–3	30.0–27.5	[6,43,56]
Cotton	1.6–1.5	310–191	12–5	8.0–7.0	[8,56,59]
Sisal	1.2–1.5	430–335	22–9	8.0–2.5	[6,10,59]
Kenaf	1.5–1.4	930–641	53–36	1.6–3.5	[55,61,62]
Bamboo	1.1–0.6	140–600	89–48	_	[8,61,63]

**Table 4 polymers-13-03612-t004:** Common chemical and physical treatments used on natural fibers.

Natural Fibers	Treatments Used	Author, Year
Chemicals		
Bonga Hemp Kenaf Papaya Pineapple Sisal Jute	Alkaline NaOH; KOH, LiOH	[8,50,68,69,70,71]
Rubber Sugar cane Kenaf Jamaica flower Papaya Ramie Hemp	Siliano SiH_4_ (3-glycidyloxypropyl) trimethoxysilane	[4,6,10,68,72,73,74]
Sisal	Acetylation CH_3_COOH	[1,43]
Coconut Sugar cane Cereal straw	Hydrogen peroxide H_2_O_2_	[6,63,75]
Jute	Benzoylation C_7_H_5_ClO	[6,63,76]
Bamboo Hemp Trupillo Sisal	Potassium permanganate KMnO_4_	[4,43,60,77]
Sisal Jute	Stearic acid	[72,78]
Physical		
Fique, jute, coconut, linen and cotton	Surface fibrillation	[42]
Wheat, rye, coconut	Cold plasma Electric shocks Crown	[9,79,80]
Ramie, falx, Kenaf and jute	Steam blast	[9]

**Table 5 polymers-13-03612-t005:** Some applications of the polymeric matrix compounds reinforced with natural fibers.

Areas	Applications	Author, Year
Automotive and aeronautics	Interior components, windshields, bumper packaging, brake pads, head restraints.	[4,5,8,18]
Insulating materials	Sound insulation, thermal insulation, spill insulation, blowing insulation, panels, acoustic soundproofing.	[6]
Architecture	Sunscreens, walls, coatings, decorative materials, sanitary ware, chairs and tables, kitchen sinks, tile, paper, floors.	[63,95]
Marine environments	Marine docks, boats and their interior components (door panels, cockpit pillars, parts of the seats).	[5,6]
Infrastructure and Construction	Load-bearing structures such as beams, roofs, multipurpose panels, pedestrian bridges, bricks, and low-cost housing.	[5,49,65,87]
Biomedical	Restoration of teeth, medical tubes, implants, administration of drugs, bone fixation plates, bone cement, and bone grafts. Vascular grafts, intraocular lenses, pacemakers, biocensors, heart valves.	[1,40,50,66]
Military	Ballistic protection, dashboards, helmets, vests, shields, shield components in vehicles.	[96,97]
Recreation and sport	Musical instrument boards, helmets, chest protectors, leg protectors, hockey sticks, balls, snowboards, fishing pole, golf clubs, swords, cleats, and skis.	[63,66,98]
Electrical and electronic	Transmission towers, isolators, and phone and laptop cases.	[2,13,33]
Textile and footwear	Garment fabric, soles, insoles, upholstery, sacks, burlap cloth, carpet strings, carpet liners, and twine.	[9,11,18,63]

## Data Availability

The data presented in this study are available on request from the corresponding author.

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
