# Peer review of "Recycling of Residual Polymers Reinforced with Natural Fibers as a Sustainable Alternative: A Review"

_polymers, 2021, doi:10.3390/polym13213612_

Round 1

Reviewer 1 Report

Dear Author

thank you for your work, but I don't understand the relationship between title and content!!!!!

the title needs to change because it is not recycling, as you mentioned in the abstract,

the title can be changed to “behavior structural of natural fibers and their components in the context of engineering applications as reinforcements of the residual polymers”

furthermore, you to speak about the use of the fiber waste using a different polymer as a composite in a different field, including the characterization of the prepared composite

Author Response

Corrections to manuscript: polymers-1413202

“Recycling of residual polymers reinforced with natural fibers as a sustainable alternative: A review”

Below are the changes recommended by the referees

Referees 2. The authors want to give an overview about recycling of residual polymers reinforced with natural fibers but there is no information about recycling. I think that they need to add a paragraph about degradation, composting, reprocessing, Life-Cycle Assessment of this type of composites. The literature is very reach and current.

In addition, I ask to enrich the one already present as well. I've listed some refs below.

Authors: We carried out an analysis on the recycling of residual polymers, adding the following review to the document:

  1. Recycling of residual polymers

High production and increasing demand for polymeric materials cause considerable amounts of non-biodegradable solid waste accumulated in landfills, affecting the natural dynamics of ecosystems; recycling being one of the most successful environmental man-agement strategies with positive results that offer several application options, this section describes the technologies, the principles involved and the challenges they imply in terms of efficiency, economic viability and environmental solutions.

The recycling of residual polymers is carried out by different methods depending on the nature of the components and the application of the products obtained [19, 20, 21], the most common as evidenced in Table 1. are i. mechanical recycling through crushing and/or grinding of polymers to reduce size, for being the most appropriate, with the least limitations and friendly with the environment ii. chemical recycling converts polymers into useful monomers in the petrochemical in-dustry through solvolysis and hydrolysis and iii. thermal recycling of polymers by pyrol-ysis into new materials with energy use.

The environmental challenges of mechanical recycling of polymeric waste, It has an emerging field that is the composites reinforced with natural fibers which are described extensively in the present review; which lies in extracting natural fibers of low commercial value and easy to obtain, to reinforce the matrices of these biodegradable products with various engineering applications [22, 23].

Table 1. Recycling methods, techniques, reuse and drawbacks

Recycling method

Techniques and processes

Reuse   

Drawbacks

Authors

Mechanical

Crushing and/or grinding to reduce the size of polymer waste

Raw material to produce compounds for the same or other applications.

Limitations in its use due to the low density, interfacial adhesion and mechanical resistance of the resulting fibers.

22, 24 25

Chemical

Chemical degradation of polymeric residues into basic chemicals.

Raw materials for petrochemical industries Alternative fuels.

Water and air pollution problems, it is dangerous for the health of recyclers.

22, 26, 27

Thermal

Recovery by pyrolysis and fluidized bed.

Energy, cement production.

Problems of atmospheric pollution due to the emission of low molecular weight gases.

29, 28, 22

2.1 Mechanical recycling

Is a method to reduce the size of polymeric materials into small fragments with sizes ranging between 20 and 200mm [24], polymeric compounds are crushed, ground, reprocessed and combined with particles, depending on the interest, the resulting fractions of the process can be classified into different sizes using sieves in powders, fibers or flakes [23, 25]. They are used in the produc-tion of new products, with limitations due to their low density, mechanical strength, interfacial adhesion, so it is necessary to apply treatments and use reinforcing fibers as mentioned below.

2.2 Chemical recycling

The compounds present in the residual polymer disintegrate, recovering the monomers through the chemical depolymerization of the polymer chains [23] using different chemicals such as acids, bases and solvents depending on the nature of the polymeric substrate, when organic and inorganic solvents are used it is known as solvol-ysis and if water is used as hydrolysis [26, 27], widely used in the petrochemical industry in the development of new alternative products and as fuel, economic and high performance, with limitations due to the environmental impacts generated by the emissions and discharges of the process residues in the receiving environment.

2.3 Thermal recycling

With energy recovery, it is a process where heat is used to decompose the polymer into its different compounds, with variable temperatures depending on the type of residual poly-mer ranging between 500 and 700 °C [29, 30], the ener-gy generated in the process is used most of the time, despite the drawbacks due to the by-products of ash and low molecular weight gases that are produced with the volatiliza-tion of some polymer compounds [28].

Thermal recycling can be done by fluidized bed and pyrolysis process, which are complex but their products are potentially useful and clean, similar to virgin products [24, 31]. The fluidized bed process passes a rapid stream of hot air through silica sand to break down the composites present in polymers at low temperatures [31] and the pyrolysis process the composite materials present in the polymer are heated in the absence of oxygen and decompose producing oil, gas, fibers and fillers [20, 24].

Advances in research on the recycling of polymers reinforced with natural fibers has been responding to the growing concern of environmental management, making it an area of interest for the transformation of polymeric waste into commercially viable products; However, it is necessary to improve knowledge of recovery rates, the useful life of products, large-scale recycling, increased compatibility with the environment, and that they are so-cially safe.

Referees 2. Pag 2 line 53.  I suggest to the authors to describe also the principal drawbacks of these class of materials. I think that some information can be obtain from this review:

Mazzanti, V., Mollica, F. A review of wood polymer composites rheology and its implications for processing Polymers, 2020, 12(10), pp. 1–23, 2304

Authors: some inconveniences are added in the elaboration of polymeric matrices on pag 2 line 53.

Natural fibers are flexible filaments, extracted from renewable sources, with complex properties, due to the wide variations in the chemical and structural composition of cellulose, lignin, hemicellulose, fatty acids, pectins and other structures linked by intermolecular hydrogen bonds and forces of Van der Waals forming microfibrils with parallel ar-rangement; They are used in different fields of engineering as a reinforcing element that provides resistance to tension and bending, rigidity, modulus of elasticity [13, 19, 20]; forming a remarkable bioengineering material of interest for its multiple uses and appli-cations, in a wide range of products with pleasing value, which differs from the initial materials, relating it to the nature of the fibers, crystallinity and insolubility; but the elaboration of these composite materials in the melt process can present drawbacks such as the viscosity, which can become very high, especially when the fiber content is greater than 50% by weight; Mazzanti, et al., [34] in their research says that if the ca-pillary is too narrow, the pressure can become too high and the use of lubricants promotes the sliding of the wall, which can alter the viscosity measurements if it is not necessary. account for wall slip appropriately. [35, 36].

Referees 2. Pag 4 line 126-127 Here the authors can add some references about papers that describe natural fibers benefits for example:

Acoustic absorption: Santoni et al. Characterization and vibro-acoustic modeling of wood composite panels Materials, 2020, 13(8), 1897

Thermal insulation: Abedom F. et al. Development of Natural Fiber Hybrid Composites Using Sugarcane Bagasse and Bamboo Charcoal for Automotive Thermal Insulation Materials Advances in Materials Science and Engineering Volume 2021 Article number 2508840

Authors: some references are added to articles that describe the benefits of natural fibers in compounds used for acoustic absorption and thermal insulation on the pag 4 line 126-127.

Natural fibers when used as reinforcements in biocomposites, in addition to repre-senting environmental benefits, reduction in energy consumption, insulation properties and acoustic absorption [53. 54], also have essential mechanical properties, as evidenced in Table 32, with average variations ranging from 1.25 - 1.5 g/cm3 for density, 320 - 520 Mpa for tensile strength, 22 - 48 Gpa for tensile mod-ulus and 7 – 25 % for elongation before break, additionally, the investigations of Nurazzi et al., [10]; Nagaraj et al. [37] and others relate flexural strength, modulus of elasticity, thickness swelling, and water absorption as important properties when evaluating them when they are used as polymeric reinforcements.

Referees 2. Pag 7 line 257 There are specific papers about impact resistance and toughening agent.

For example:

Correlation between mechanical properties and processing conditions in rubber-toughened wood polymer composites Mazzanti V. Polymers, 2020, 12(5), 278

Mechanical properties and morphology of impact modified polypropylene-wood flour composites Oksman, K. (1998) Journal of Applied Polymer Science, 67 (9), pp. 1503-1513

Toughening of wood-plastic composites based on silane/peroxide macro crosslink poly(propylene) systems Meekum U. (2018) BioResources, 13 (1), pp. 1678-1695.

Authors: a paragraph is added that describes the influence of hardeners on the impact resistance of composite materials, pag 7 line 257.

To improve impact resistance, hardeners or impact modifiers are used in the production of biocomposites. The investigations of Khui, et al., [91]; Meekum and Khongrit., [92] used hardeners to improve the interfacial bond between fiber-matrix, finding an increase in the values of impact resistance and elongation at break. These addictives are not only used as a hardener, but also as compatibilizers that increase the tensile properties of materials [93].

Mazzanti, et al., [94] in their research they use rubber as a hardener in the elaboration of polymeric matrices based on polypropylene reinforced with wood, with different percentages of addiction and processing method; finding that to provide a significant improvement in the toughness and ductility of the compound a minimum amount of 20% by weight of hardener is required.

Referees 1: thank you for your work, but I don't understand the relationship between title and content!!!!!

The title needs to change because it is not recycling, as you mentioned in the abstract,

The title can be changed to “behavior structural of natural fibers and their components in the context of engineering applications as reinforcements of the residual polymers”

furthermore, you to speak about the use of the fiber waste using a different polymer as a composite in a different field, including the characterization of the prepared composite

Authors: Based on the recommendations of the two reviewers, we decided not to change the title of the manuscript and include a description about the recycling of residual polymers that relates the title and content.

Authors: All references changed order

Reviewer 2 Report

The review paper “Recycling of residual polymers reinforced with natural fibers as a sustainable alternative: A review” is quite interesting and centered on the scope of the journal. The paper is suitable to publish but a new important part need to be added.

The authors want to give an overview about recycling of residual polymers reinforced with natural fibers but there is no information about recycling. I think that they need to add a paragraph about degradation, composting, reprocessing, Life-Cycle Assessment of this type of composites. The literature is very reach and current.

In addition, I ask to enrich the one already present as well. I've listed some refs below.

Pag 2 line 53  I suggest to the authors to describe also the principal drawbacks of these class of materials. I think that some information can be obtain from this review:

Mazzanti, V., Mollica, F. A review of wood polymer composites rheology and its implications for processing Polymers, 2020, 12(10), pp. 1–23, 2304

Pag 4 line 126-127 Here the authors can add some references about papers that describe natural fibers benefits for example:

Acoustic absorption: Santoni et al. Characterization and vibro-acoustic modeling of wood composite panels Materials, 2020, 13(8), 1897

Thermal insulation: Abedom F. et al. Development of Natural Fiber Hybrid Composites Using Sugarcane Bagasse and Bamboo Charcoal for Automotive Thermal Insulation Materials Advances in Materials Science and Engineering Volume 2021 Article number 2508840

Pag 7 line 257 There are specific papers about impact resistance and toughening agent.

For example:

Correlation between mechanical properties and processing conditions in rubber-toughened wood polymer composites Mazzanti V. Polymers, 2020, 12(5), 278

Mechanical properties and morphology of impact modified polypropylene-wood flour composites Oksman, K. (1998) Journal of Applied Polymer Science, 67 (9), pp. 1503-1513

Toughening of wood-plastic composites based on silane/peroxide macro crosslink poly(propylene) systems Meekum U. (2018) BioResources, 13 (1), pp. 1678-1695.

Tab 4 is in spanish i think that it needs to be translate in english

Author Response

Through this message, the corrections made to the manuscript are sent, according to the revisions

Round 2

Reviewer 2 Report

The authors have implemented the suggestions accordingly. The paper is ready to be published.